# Assessing the Impact of Science in the Implementation of the United Nations Convention to Combat Desertification

Mariam Akhtar-Schuster [1], Lindsay C. Stringer [2,*], Graciela Metternicht [3], Nichole N. Barger [4], Jean-Luc Chotte [5] and German Kust [6]

1   DLR Projektträger, Sachsendamm 61, 10829 Berlin, Germany; mariam.akhtar-schuster@dlr.de
2   Department of Environment and Geography, University of York, York YO10 5DD, UK
3   Earth and Sustainability Science Research Centre, University of New South Wales, Sydney, NSW 2052, Australia; g.metternicht@unsw.edu.au
4   Department of Ecology and Evolutionary Biology, University of Colorado, Boulder, CO 80309, USA; nichole.barger@colorado.edu
5   French National Institute Research for Sustainable Development, 34394 Montpellier, France; jean-luc.chotte@ird.fr
6   Institute of Geography, Russian Academy of Sciences, Staromonetny Lane 29, 119017 Moscow, Russia; gkust@igras.ru
*   Correspondence: lindsay.stringer@york.ac.uk

**Abstract:** In 2013, the United Nations Convention to Combat Desertification (UNCCD) established a science–policy interface (SPI) to address Parties' need for demand-driven, timely, interdisciplinary science and technical knowledge to tackle problems of desertification, land degradation and drought. Since then, a comprehensive assessment of the SPI's impacts on policy decision-making has been lacking, despite perceptions that the SPI is vital to the Convention's success. Addressing this gap, this paper evaluates whether the SPI and its processes and outputs have provided the necessary scientific and technological knowledge and advice to Parties to support timely, evidence-informed decision-making. It applies an analytical framework to assess performance metrics, considering associated documents and evidence of societal relevance and social quality. The findings indicate that SPI outputs have improved implementation of the UNCCD since 2015, particularly in the context of Sustainable Development Goal Target 15.3. SPI outputs have supported scientific cooperation between the Convention and its strategic partners while enhancing its science and technology profile in line with Article 16 and Article 17. The findings indicate that further formalization of the SPI's status within the UNCCD is vital to improve its functions, undertake its work, and enable the UNCCD to maintain its global lead in providing knowledge and advice on combating desertification, land degradation and drought.

**Keywords:** science–policy interface; desertification; land degradation; sustainable development goals (SDGs); knowledge management; UNCCD; sustainable land management

## 1. Introduction

The United Nations Convention to Combat Desertification (UNCCD) is the sole legally binding multi-lateral environmental agreement tackling challenges of desertification, land degradation and drought from both an environmental and developmental standpoint. While the UNCCD specifically addresses dryland areas, it fosters sustainable land management across all climatic zones, while regional annexes allow these challenges to be addressed according to the specific regional contexts in which they occur. The UNCCD encourages Parties to "*adopt an integrated approach for addressing the physical, biological and socio-economic aspects of the processes of desertification and drought*" (Article 4) [1]. It supports international cooperation in the fields of technology transfer, as well as scientific research and development, information collection and dissemination. Two articles of the

Convention directly target scientific and technical cooperation focusing on information collection, analysis and exchange (Article 16), and research and development (Article 17). Article 16 specifies that "*[T]he Parties agree, according to their respective capabilities, to integrate and coordinate the collection, analysis and exchange of relevant short term and long term data and information to ensure systematic observation of land degradation in affected areas and to understand better and assess the processes and effects of drought and desertification*" [1]. Article 17 of the Convention establishes that "*[T]he Parties undertake, according to their respective capabilities, to promote technical and scientific cooperation in the fields of combating desertification and mitigating the effects of drought through appropriate national, subregional, regional and international institutions*" [1].

Although Articles 16 and 17 emphasize the importance of science, research, technology and development for "achieving improved productivity as well as sustainable use and management of resources . . . . that improve the living standards of people in affected areas" [1] the UNCCD's original design lacked a consolidated mechanism to channel scientific inputs to inform its implementation. While the Committee on Science and Technology (CST) (see Article 24, [1]) is a key subsidiary body of the Convention, it serves a political rather than scientific role, and comprises mostly governmental representatives [2]. This challenge has historically restricted the Convention's ability to play a leadership role in providing knowledge and advice on desertification, land degradation and drought [2–4].

Years of discussions involving policymakers, scientists and civil society organizations (CSOs) [5–10] supported calls to help the CST to "*accelerate its efforts to establish links with scientific communities*" [10]. In 2007, Parties urged the CST to organize each of its future sessions in a "*predominantly scientific and technical conference-style format*" [10]. Findings from the first conference-style meeting in 2009 highlighted the need for an "*independent, international, interdisciplinary scientific advisory mechanism*" [11] to support progress towards Articles 16 and 17 [1]. A flexible "modular mechanism" was proposed, whereby work is driven by the demands of the Parties to the UNCCD for science and technology knowledge, comprising: (i) a science–policy interface (SPI); (ii) an international self-governing and self-organizing independent group of scientists; and (iii) Regional and Science and Technology Hubs in each UNCCD region [12,13]. At the CST's 11th session, Parties established the SPI to "*facilitate a two-way science-policy dialogue and ensure delivery of policy-relevant information, knowledge and advice on desertification/land degradation and drought*" [14]. Independent scientific advice for this new committee was to be provided through the CST's selection of ten SPI members based on a robust analysis of experts' scientific qualifications and science-based competencies. The resultant information flows through the UNCCD architecture are shown in Figure S1 in the Supplementary Materials. A list of acronyms and abbreviations used throughout this paper is also provided (List S1).

By Decision 23/COP.12, three years after commencing its work, the SPI underwent an independent external review [15] focusing on "*the work conducted by the SPI during the biennium 2016–2017 and [ . . . ] its overall achievements since its establishment in order to decide on the future functioning of the SPI*" [16]. The review concluded that "*the SPI has made a promising start*" [15]. However, it highlighted resource limitations, a mismatch between available resources and the volume of work expected, and the need for better integration of SPI members into the implementation of its future work programs. The review further noted the need for greater visibility of the SPI's work to improve the recognition and consideration of its findings among the UNCCD's partners and the scientific community.

Despite these findings, there has been no assessment of the impact of the SPI in the UNCCD process and whether it has helped guide progress towards the Convention's overall goals. This paper addresses this knowledge gap. Our objectives are, therefore, to:

- Assess the SPI's impact on the UNCCD process; and
- Evaluate whether the SPI contributes to the achievement of the objectives of the Convention outlined in Articles 16 and 17, and thus UNCCD implementation.

Our findings could assist the UNCCD's future planning, supporting evidence-informed decisions and actions in tackling desertification, land degradation and drought.

## 2. Materials and Methods

To assess the SPI's impact on the UNCCD process (Objective 1), we explored how the SPI's outputs are: (a) directly entering the UNCCD to support evidence-informed decisions and actions of Parties to the Convention, (b) strengthening interactions between the UNCCD and external stakeholders and partners, and (c) enabling the Convention to substantially contribute to the realization of the UN 2030 Agenda for Sustainable Development and other relevant international agreements. We considered the SPI's mandate and the processes that direct its work, using this to determine criteria to systematically analyze UNCCD documents, decisions, SPI technical reports, policy briefs and other products from 2013 to 2019. Four criteria identified as essential to assess points (a–c) were: (1) policy and thus societal relevance of SPI products in the context of the Convention and their impact on decisions, (2) evidence of the promotion of technical and scientific cooperation to strengthen scientific and technical cooperation with UNCCD strategic partners, (3) evidence that accounts for the underlying participatory principle that guide the Convention, and (4) evidence of the impact of SPI beyond COP actions (see Table S1).

Based on the Framework for the Quality Assessment of Social Science Research (Royal Netherlands Academy of Arts and Sciences, 2013), we assessed the use of SPI outputs in support of evidence-informed decision-making in Section 3 (Figure 1). The *Framework for the Quality Assessment of Social Science Research* [17] fulfilled the four criteria outlined above and guided our assemblage of indicators focused on the quality domain of "societal relevance" as related to our objectives (see Table S2 for details on the framework used). Subsequently, findings from Objective 1 were mapped onto the clauses under Articles 16 and 17, to address Objective 2 (Tables S2 and S8).

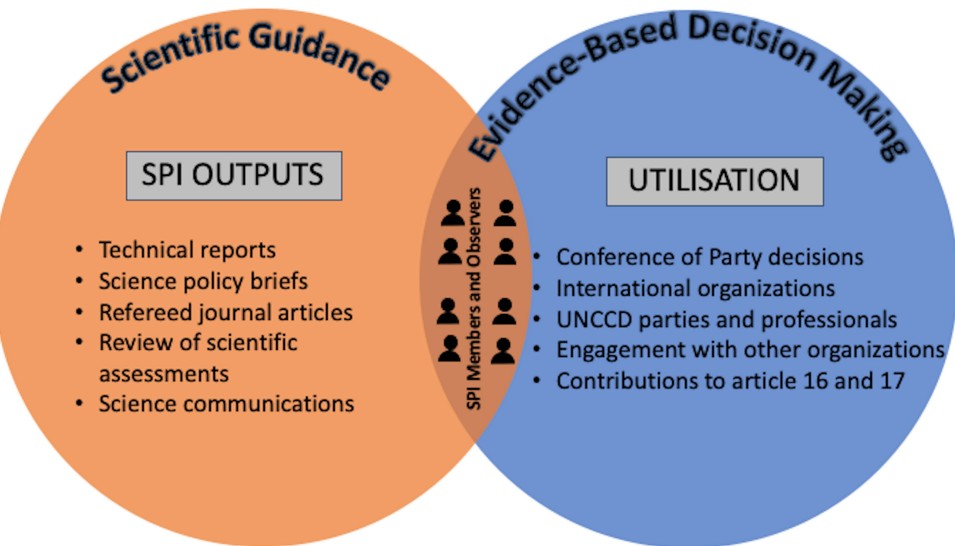

**Figure 1.** Framework to analyze SPI outputs and subsequent utilization by decision makers within the UNCCD context. The framework for assessing the utilization of SPI outputs was adapted to the assessment dimensions and quality indicators outlined in the Framework for the Quality Assessment of Social Science Research [17].

A mixed-method approach [18] was adopted to analyze official, publicly available outputs that emerged from the three successive SPI biennial work programs, the associated documents of the CST, the Convention Implementation Review Committee (CRIC), and the Global Mechanism (GM), as well as the resulting decisions and other actions of the UNCCD COP. Online searches of ongoing initiatives of UNCCD and its strategic partners were used to: (a) identify and analyze the SPI's demonstrated impact on UNCCD strategic partnerships and (b) assess the SPI's engagement with other organizations and the use of its products by professionals. Personal communications were also conducted with staff of

the UNCCD Secretariat, the CRIC and the GM, and past and current SPI members to gain important background context. Information triangulation from these different sources was applied to converge evidence (see Table S2 for more details on the methods used).

One core aspect of the research was to analyze 98 Land Degradation Neutrality (LDN) national target-setting reports—produced by countries with the support of the UNCCD GM—against a set of direct and indirect criteria derived from the Land Degradation Neutrality Conceptual Framework (Conceptual Framework for LDN) [19,20] to ascertain whether SPI science advice underpins LDN national target-setting activities (see Tables S2 and S6a,b for details).

Noteworthy is that for the very early work of the SPI it was difficult to trace its contribution as a scientific source (e.g., adoption, endorsement, or promotion of its proposals by a COP), as the information was implemented as a matter of course in various other areas of UNCCD work over the years. This may have led to some gaps in capturing the full scope of SPI inputs to UNCCD COP decisions and related activities. The analyses enabled us to identify barriers that adversely impact performance and to uncover untapped or under-utilized opportunities.

### 3. Results

*3.1. SPI's Contributions to Decisions of the UNCCD Conference of the Parties*

Several mechanisms facilitated a "two-way science-policy dialogue to ensure delivery of policy-relevant information, knowledge and advice on desertification/land degradation and drought" [14] by the SPI, providing a frame that supported the uptake of the SPI's findings in COP decisions (Figure 2). Mechanisms include: (a) presentation and discussions of SPI findings at the CST plenaries in 2015, 2017 and 2019; (b) participation in contact groups and meetings of the CST Bureau to provide scientific advice in preparation for decisions and related actions for the high-level segment of the COP; (c) presentations at side events in advance of the high-level segment, providing further information required by Parties and other relevant stakeholders; (d) SPI meetings at the COP venue that enabled strategic internal SPI-related and external communication with Parties and strategic partners during the ongoing negotiations; (e) the availability of SPI members during the entire COP in case Parties required further ad hoc consultations; and (f) organization of the first UNCCD Science Day that supported science–policy and science–science dialogue (Table S7c).

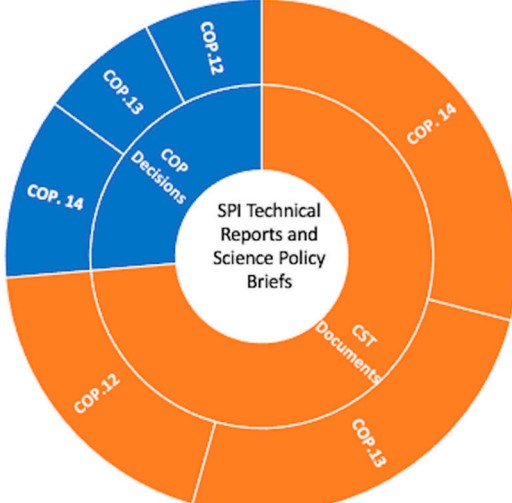

**Figure 2.** Scientific input from the SPI into COP decisions. Proportion of SPI technical reports and science policy briefs that were incorporated into CST documents and COP decisions over three COPs from 2012 to 2019.

### 3.1.1. SPI's First Biennial Work Program (2014–2015), Addressing Decisions Made at UNCCD COP.12

Five out of six CST work documents were directly informed by products of the 2014–2015 SPI Work Program. Four out of six CST information documents appear directly influenced by the outcomes of this first SPI Work Program, contributing to 9 of the 35 decisions taken at UNCCD COP.12 (Figure 2; Table S3a).

During this period, the SPI also addressed how to best measure progress on strategic objectives 1, 2 and 3 of the UNCCD's 2018–2030 Strategic Framework [21], and brought scientific evidence to the other Rio Conventions on the contribution of sustainable land management to climate change adaptation/mitigation and to safeguarding biodiversity and ecosystem services. The SPI contributed to the UNCCD's efforts to identify indicators to understand the status of land. This information supported the CST Bureau in exploring options for further harmonization of progress indicators, considering how they can be used in the other Rio Conventions to generate synergies and facilitate reporting. According to the CRIC, the process of using progress indicators for joint monitoring by UNCCD, the Convention on Biological Diversity (CBD) and the United Nations Framework Convention on Climate Change (UNFCCC) was "*largely conducted by the SPI*" [22].

Decision 23/COP.11 also mandated the SPI to "analyze, synthesize and translate relevant scientific findings and recommendations from desertification/land degradation and drought-related scientific conferences, including upcoming UNCCD scientific conferences" [14], to seek ways to increase the effectiveness of UNCCD scientific conferences in providing policy-relevant information, knowledge and recommendations. CST recommendations informed by the SPI were adopted in Decision 19/COP.12, leading to a substantial reorganization of the way science entered the UNCCD and expanded the mandate of the SPI (see Decision 23/COP.11). Consequently, the SPI provided thematic guidance to the CST on scientific knowledge requirements, and under its subsequent work programs, to "identify the most optimal way forward" [16] to meet the knowledge requirements of the UNCCD. As early as 2015, the CRIC noted that "[T]he work carried out under the CST linked up with the work of scientific organizations and cooperation bodies that deal with issues relevant to DLDD [desertification/land degradation and drought] perhaps more than ever before. The establishment of the SPI was an important means in this regard" [22]. Successful implementation of the SPI's first work program is also reflected by Decision 10/COP.12 [16], which earmarked resources within the UNCCD Trust Fund for the biennium 2016–2017 for SPI activities.

### 3.1.2. SPI's Second Biennial Work Program (2016–2017), Addressing Decisions Made at UNCCD COP.13

The SPI's second work program focused entirely on scientific inputs to the CST. Seven of eight CST working documents were directly informed by activities and products of the 2016–2017 SPI work program, as were both CST information documents. Together, these contributed to 9 of the 36 decisions taken at UNCCD COP.13 (Figure 2; see also Table S3b). As a result of procedural advances in its operation, by 2016–2017 the SPI developed its own technical reports and associated policy briefs, rather than directly contributing to the development of the CST documents. This allowed greater visibility for the SPI's scientific and technical contributions, with its outputs using a standard format and logo. SPI technical reports were published online and in hard copy in preparation for COP negotiations following internal and international independent peer review in line with Decision 19/COP.12 [16].

The 12th COP (Decision 3/COP.12, UNCCD, 2016) requested the development of guidance to formulate national Land Degradation Neutrality targets (LDN), to support operationalization of the voluntary SDG LDN target 15.3. The SPI, therefore, developed a scientific Conceptual Framework for LDN to provide practical guidance for countries, including ways to monitor national progress towards the LDN target, which was endorsed at the UNCCD COP.13 by Decision 18/COP.13 (Table S3b). Section 3.3 discusses the multi-

farious and sustained impacts of the Conceptual Framework for LDN after its endorsement. Decision 2/COP.13 also invited Parties to use the concept of LDN to foster coherence among national policies, actions, and commitments.

The COP also sought to promote and facilitate the adoption of sustainable land management practices, requesting the SPI to assess the synergistic potential of these practices to address "*desertification/land degradation and drought, climate change mitigation, and climate change adaptation*". The technical report addressing this request informed CST recommendations at UNCCD COP.13 in 2017. It also led to Decision 18/COP.13, which called on Parties to consider the use of locally adapted sustainable land management practices as an effective means of achieving land-based national objectives related to addressing desertification, land degradation and drought, LDN, and in improving livelihoods and socio-economic conditions (Table S3b).

The COP also requested the SPI's advice on rehabilitation, restoration and reclamation measures and practices in degraded areas (Decision 18/COP.13) and more collaboration with the International Resource Panel of the UN Environment Program (preparing a report on land restoration and the SDGs, with a focus on SDG Target 15.3). This suggests that, as early as 2017, Parties began to perceive the SPI as key to further develop and expand the UNCCD's profile as a scientific authority on SDG 15.3 (Table S3b).

### 3.1.3. SPI's Third Biennial Work Program (2018–2019), Addressing Decisions Made at UNCCD COP 14

All working documents the CST developed for COP.14 were directly informed by the activities and products of the SPI's work and contributed to 13 of the 33 decisions adopted at the 14th COP of the UNCCD (Figure 2; Table S3c).

The SPI provided additional guidance for the implementation of LDN (Table S3c), developing a technical report that included tools for soil organic carbon (SOC) estimation in the context of planning and monitoring for LDN. The SPI technical report on aspects that support an enabling environment for LDN and its potential contribution to enhancing well-being, livelihoods, and the environment informed CST recommendations to UNCCD COP.14 on science-based evidence on integrated land use planning, governance and institutional development for LDN. The SPI also contributed to a synthesis report of the UNCCD on relevant case studies for LDN implementation (Decision 13/COP.14, [23]).

Parties requested the SPI to address the land-drought nexus concept, an aspect included in the mandate of the UNCCD but lacking concrete and practice-oriented science on how to tackle it through land-based interventions. The SPI developed the concept of drought-smart land management to guide actions that mitigate drought effects, in tandem proactively managing drought risk to improve food and water security (Table S3c). Parties were invited to support the adoption and implementation of land-based interventions for drought management and mitigation. Parties also requested the SPI to work with the FAO, UNEP, WMO and other relevant organizations in the context of an Integrated Drought Management Program (Decision 17/COP.14, [23]) to develop a shared understanding of this phenomenon (e.g., shared definitions) (Table S3c).

The results of SPI's work on refining and using UNCCD progress indicators for setting and implementing LDN targets under the Convention influenced the COP's decisions as outlined in the statement: "*Another focus area, for which the work was largely conducted by the SPI, was to consider the use of the UNCCD progress indicators for joint monitoring among the three Rio conventions*" (ICCD/CRIC(14)/3:13, [22]). At COP.14, the CRIC also reiterated that "*[T]he work carried out under the CST linked up with the work of scientific organizations and cooperation bodies that deal with issues relevant to DLDD perhaps more than ever before. The establishment of the SPI was an important means in this regard.*" (ICCD/CRIC(14)/3: 13, [22]).

The SPI's contributions to Decisions of the UNCCD COP in its 12th, 13th, and 14th sessions show a gradual yet progressive shift in the focus of the work. While the SPI's contributions under its first work program addressed scientific and technical procedural requirements in the context of improving pathways for science to enter the UNCCD, under

its second work program the SPI was able to focus on science guided by clearly structured work procedures. The third SPI work program focused on providing practical guidance for implementing the evidence-based science emerging from its work, providing knowledge inputs, increasingly in collaboration with strategic partners of the UNCCD.

### 3.2. The Role of the SPI in Furthering UNCCD Strategic Partnership-Building

All biennial work programs to date have requested the SPI support the Bureau of the CST in initializing and further developing strategic partnerships with IPBES, Intergovernmental Technical Panel on Soils (ITPS) and IPCC, as originally requested in Decision 23/COP.11 (Table S3a). The results of our analyses show that entry points for collaboration had to be identified by the SPI, considering the mandates, timetables, and planned and ongoing work of each body (Table S4a–c). Collaboration with IPBES, ITPS and IPCC consolidated and further developed, with knowledge and information sharing appearing to be mutually beneficial in each case (Figure 3). For example, the IPBES Land Degradation and Restoration Assessment, adopted by the IPBES Plenary in 2018 [24], refers to work on LDN and sustainable land management conducted by the SPI, as does the IPCC Special Report on Climate Change and Land [25].

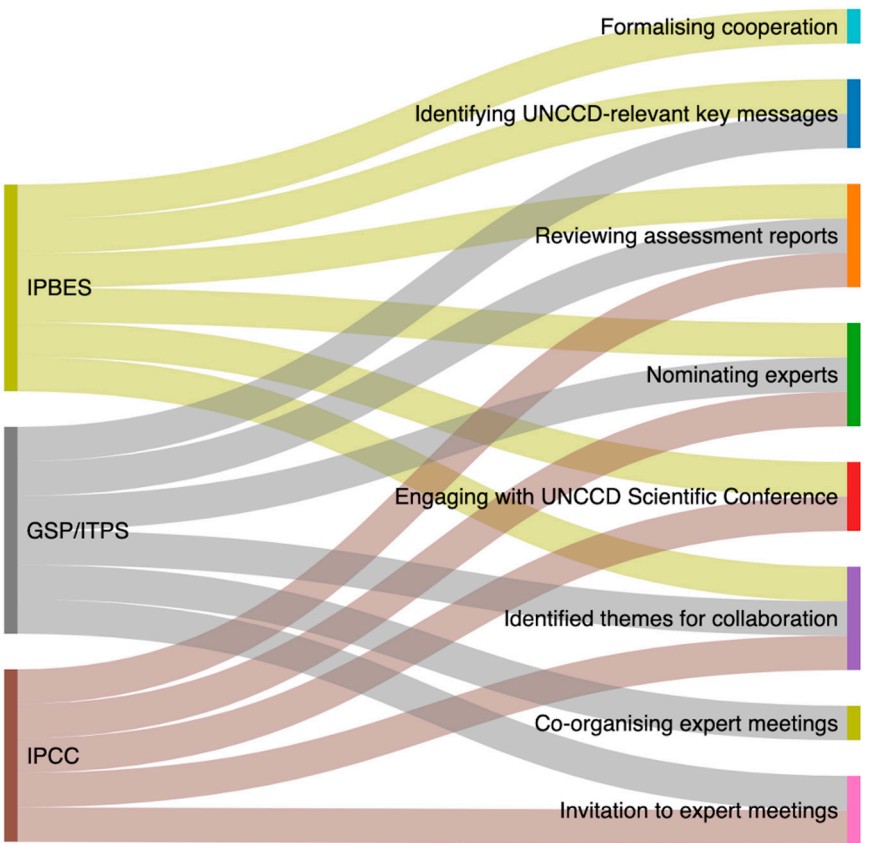

**Figure 3.** As part of its coordination activities, SPI collaborated with IPBES, IPCC, and GSP/ITPS in various ways during the three biennial work programs (WP 2014–2015; WP 2016–2017; WP 2018–2019) to identify and make available UNCCD-relevant information on desertification/land degradation and drought provided by these bodies (for details, see Tables S3a–c and S4a–c).

Work of the SPI has also increased awareness of the UNCCD and its mandate among leading UN agencies on food, agriculture and environment, as evidenced by invitations to co-organize the Global Symposiums on Soil Organic Carbon (2017) (Table S4b), and on Soil Erosion (2019) (Table S4c) under the leadership of FAO, contributions to the development of the Think Piece on Land Restoration for Achieving the SDGs by the International Resources Panel (IRP) of the UN Environment Program (Table S4c), and initiation of cooperation with

the Global Land Indicators Initiative (GLII) (Table S4c). The SPI also assisted Parties to the Convention on aspects of gender mainstreaming, as stated in Decision 24/COP.14 [26]. Each of these issues is highly pertinent to addressing DLDD, and the work of the SPI enabled important new contributions within each of these linked areas.

These collaborations provide evidence of increased awareness and regard for the work of the SPI concerning: (a) advancing the UNCCD's strategic partnership-building with international organizations, and strengthening UNCCD's scientific expertise on desertification, land degradation and drought, as well as SDG target 15.3; (b) ensuring that assessments conducted by other science–policy interfaces remain relevant to the UNCCD. The SPI's work to consolidate strategic partnerships of the UNCCD is, however, best demonstrated by the support it provided in establishing a Memorandum of Cooperation between the secretariats of the IPBES and the UNCCD [27].

### 3.3. Utilization of SPI Products by UNCCD Parties and by Professionals

Tables S5 and S6a feature the demonstrable utilization of SPI outputs by UNCCD Parties and by professionals in the UN policy arena (UNCCD strategic partners). These tables provide evidence supporting the discussion that follows on the scientific impact of the SPI on national voluntary LDN Target Setting Projects.

Decisions 3/COP.14 and 13/COP.14 (Table S3c) requested the Global Mechanism and the secretariat of the UNCCD "to continue supporting the process of voluntary land degradation neutrality (LDN) target-setting . . . " and " . . . *to support country-level activities towards the implementation of voluntary LDN targets, including through supporting LDN transformative projects and programs*" [28]. By October 2020, a total of 102 countries had set, and technically validated, their voluntary LDN targets and associated measures, and 66 countries had formally adopted LDN targets [28]. Analysis of LDN reports from 98 countries (available on the UNCCD Knowledge Hub (status 28 April 2021) (Figure 4; see Table S6a for details) shows that all nations use the LDN concept, and to different extents, all the countries analyzed make direct or indirect mention of the conceptual framework for LDN developed by the SPI.

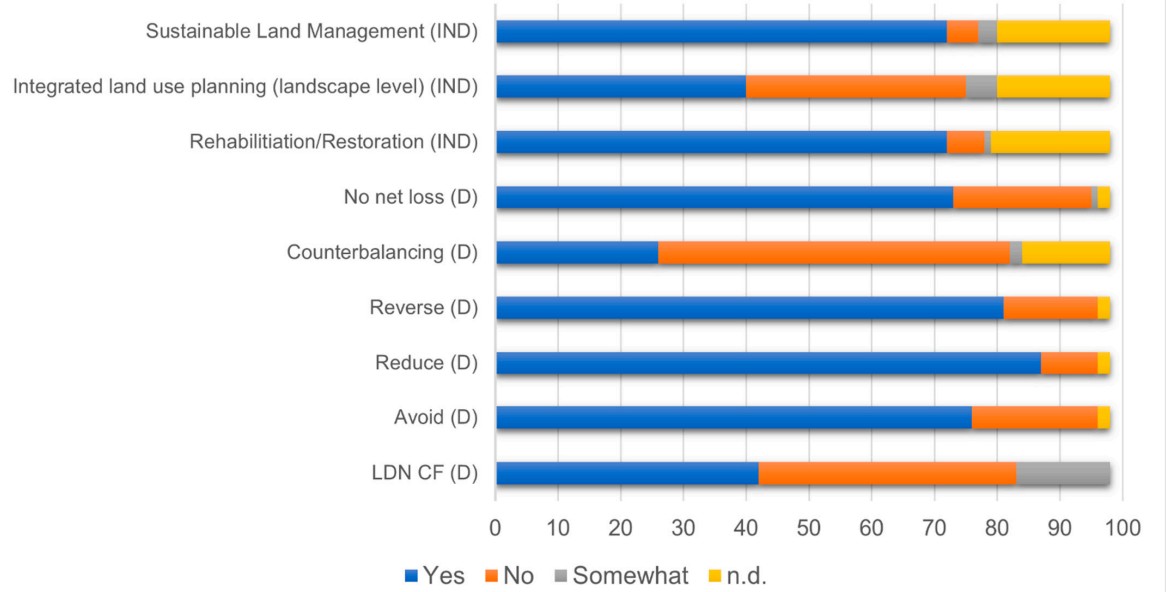

**Figure 4.** The result of screening 98 UNCCD GM national target reports using the direct (D) criteria applied in the scientific CF for LDN: LDN CF, response hierarchy (avoid, reduce and reverse), counterbalancing anticipated losses (offsetting expected losses/like-for-like approach, no net loss (neutrality), and the indirect (IND) criteria (rehabilitation/restoration, integrated land use planning, and sustainable land management) (see Table S6a,b for details); n.d. = no data.

National target setting reports for LDN (in English, French, Spanish, Russian, Arabic and Portuguese) were screened against direct and indirect criteria related to the framework (see Table S6a,b for details). Most reports suggest that countries engaged in LDN Target Setting Projects have a basic understanding of the LDN response hierarchy (avoid, reduce, reverse land degradation). All countries defined pathways to identify and operationalize a numerical baseline (or have advanced discussions about them) including policy measures and programs to enable progress assessment toward LDN.

Use of the three global indicators (land cover, land productivity and carbon stocks above and below ground (metric: soil organic carbon)) using default data provided by the UNCCD appears common practice for monitoring progress towards LDN. Metrics of these indicators were sometimes compared to existing national data. Though the response hierarchy of the Conceptual Framework for LDN is mentioned, hardly any of the reports analyzed applied the 'modules' of the Conceptual Framework for LDN [19] as part of the planning processes: whereas "vision and objectives" (module A) and "frame of reference" (Module B) are mentioned comprehensively, the "mechanism for neutrality" (module C) is scarcely addressed. This may be because of a lack of data of the necessary quality. Regarding "elements necessary to achieve LDN" (module D), c.80% of countries refer to avoiding land degradation but fail to provide specific details on how this is to be achieved. The reports show countries focused mainly on developing actions to reduce or reverse land degradation, though some did not differentiate between these. This is despite programs such as the Economics of Land Degradation Initiative highlighting the gains and effectiveness of avoiding land degradation in the first place [29]. Lastly, most reports fail to provide practical details regarding "monitoring LDN" (module E).

Some country reports strongly emphasize the integration of LDN interventions with existing National Land Use Plans or strategies. This approach may reinforce and enhance the efficacy and efficiency of LDN implementation processes, by blending and articulating LDN initiatives with other national sustainable development strategies. This suggests that through self-determined national synergistic processes, the SPI's Conceptual Framework for LDN could help attain other national and international sustainable development commitments, including biodiversity and climate change. However, the absence of a centralized national inventory system to map such co-benefits (as suggested in [20]), impedes making direct attributions to the Conceptual Framework for LDN developed by the SPI.

The indirect criteria selected for the Conceptual Framework for LDN (see Table S6a,b and Figure 4) show that national land restoration and/or rehabilitation actions fail to specify their contributions towards achieving LDN, and this becomes a drawback to mapping impact. Furthermore, most countries use the terms rehabilitation and restoration interchangeably, and it is unclear whether definitions adopted align with those of the Conceptual Framework for LDN [20]. Regarding the indirect criteria on integrated land use planning, many countries use the concept of 'watershed' or 'integrated land management' or 'integrated land use sustainability plan' as the 'landscape level'. Although all reports describe the national legal and institutional frameworks, better articulation and strengthening of their interface is needed.

These findings suggest that science synthesized by the SPI informs the implementation of the Target Setting Projects. However, the uptake of science provided by the SPI could be enhanced with complementary capacity building at the national level. For example, the use of default data (provided by the UNCCD) plays a major role in LDN target setting. This implies that preset values are used by Parties given the lack of nationally produced data, and that Parties lack information to ascertain a realistic ultimate LDN goal to be achieved. Yet, the need to strengthen the local capacity for data acquisition is not addressed in the reports.

Of importance is how outputs of the SPI have influenced the programmatic directions of mechanisms financing implementation of the UNCCD mandate, such as the Global Environment Facility (GEF). The GEF's 7th replenishment (2018–2022) continues to have land degradation as one of its focal areas, seeking to create an enabling environment for supporting countries in their voluntary LDN target implementation and to enhance the

ground implementation of sustainable land management using LDN tools. Two flagship Impact Programs of the GEF-7, the 'sustainable forest management of dryland landscapes' and the 'food, land use and restoration' programs, promote implementation of the LDN response hierarchy through their projects. The Impact Programs and land degradation focal area mention the SPI's Conceptual Framework for LDN [30]. The Conceptual Framework for LDN has, overall, strongly influenced the programmatic direction of the GEF-7 and the allocation of funds, with the land degradation focal area allocated USD 519 million [31] (11.7% of the total budget for GEF-7). Of that total, USD 25 million was allocated for enabling activities (including LDN voluntary target setting by countries), and USD 72 million was allocated to the Impact Program on sustainable forest management and dryland sustainable landscapes applying SPI-developed LDN approaches and related tools.

Furthermore, in screening projects submitted for the land degradation focal area, the Scientific Technical Advisory Panel of the GEF recommends using the LDN Transformative Projects and Programs checklist and commissioned the preparation of LDN guidelines for GEF projects [32] underpinned by the Conceptual Framework for LDN. In addition, the Food and Agriculture Organization, a GEF Implementing Agency, has directly integrated the checklist into the design of its recently launched Dryland Restoration Initiative Platform [33].

The SPI's Conceptual Framework for LDN has laid the foundation for the metadata for SDG indicator 15.3.1 [34], which underpins the global indicator framework for the SDGs and targets of the 2030 Agenda for Sustainable Development adopted by the United Nations General Assembly in 2017 [35]. The metadata document for SDG Indicator 15.3.1 states that the "conceptual framework, endorsed by the UNCCD's governing body in September 2017, underpins a universal methodology for deriving the indicator" [36]. Good Practice Guidance for SDG Indicator 15.3.1, published in 2017 [34], and reviewed and updated in 2021 [34], refers to the Conceptual Framework for LDN and its definition to calculate the extent of land degradation for reporting on SDG indicator 15.3.1. The UNCCD is now the custodian agency leading an Inter-Agency Advisory Group on 15.3.1 [34], composed of key UNCCD partners (FAO, CBD, UNFCCC and UNSD) to further refine the methodology and data tools/options for this indicator. Thus, the SPI's work on indicators has further helped to strengthen UNCCD's scientific authority on land degradation issues.

*3.4. SPI Engagement with Other Organizations and the Utilization of Its Products by Professionals beyond Its Coordination Activities*

The scientific work of the SPI is acknowledged outside the UN Agencies and Conventions. In September 2020, the G20 Ministers for Environment acknowledged the benefits of existing efforts and commitments that have also been developed for LDN and associated targets under the UNCCD and called for "increased efforts and cooperation to achieve Land Degradation Neutrality as set out in Sustainable Development Goal (SDG) 15.3 target" [37]. To support measures that avoid, reduce and reverse land degradation, the G20 launched the Global Initiative on Reducing Land Degradation and Enhancing Conservation of Terrestrial Habitats. Though no specific mention is made of the SPI's Conceptual Framework for LDN, its response hierarchy is explicitly applied, while implementation will be through an Initiative Coordination Office overseen by the UNCCD. The anticipated annual reporting of the initiative to G20 members should directly benefit from improved implementation of the voluntary LDN Target Setting Program to which 127 countries have committed. Of note is that whilst analyzing approaches for following up on the effectiveness of government interventions Pillay and Buschke [38] refer to the SPI's Conceptual Framework for LDN as a rare exception that assigns indicators to the outcome level by using indicators that are responsive to land management actions see [20,21].

The SPI's products also inspire the UNCCD's strategic partners and other professionals, being progressively adapted to the needs of these bodies without explicitly citing the SPI. For instance, the UN Decade on Ecosystem Restoration which started in 2021 reflects the

Conceptual Framework for LDN response hierarchy on its web portal by using the three-step approach of "preventing, halting and reversing the degradation of ecosystems worldwide".

Lastly, Figure 5 exemplifies SPI's engagement with other organizations beyond its coordination activities (e.g., the WMO, UNFCCC, the Society for Ecological Restoration) as outlined in Table S7c. Additional work (keynote talks, presentations, reviews, panels, juries, recorded outside of official COP decisions and not part of formal COP requests) (Table S7a–c) demonstrates that the SPI could respond flexibly to multiple opportunities relating to its work. Outcomes from these activities informed, for example, the collaboration with the Global Environment Facility—Scientific and Technical Advisory Panel (GEF–STAP).

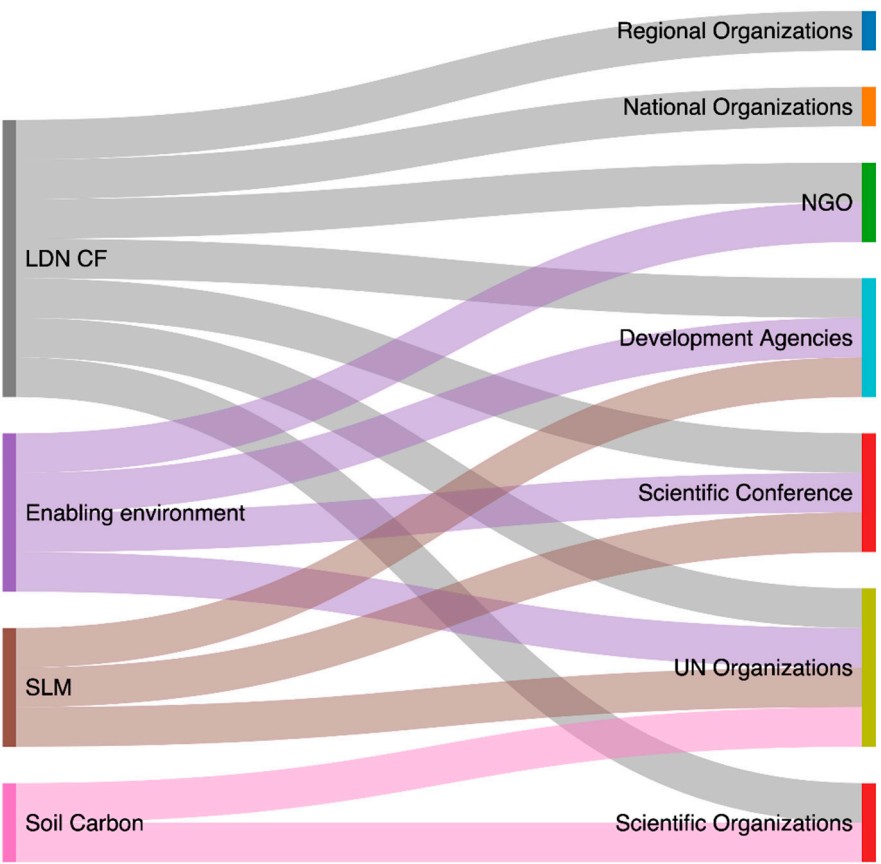

**Figure 5.** Uptake of the four SPI technical reports by various governmental and non-governmental organizations. LDN CF = [20]; Enabling environment = [39]; SLM (sustainable land management) = [40]; Soil carbon = [41]. For additional details on the uptake of these technical reports, see Table S7a–c.

*3.5. SPI's Contribution to Articles 16 and 17*

In order to address Objective 2 on whether and how the SPI is contributing to the achievement of the objectives of the Convention outlined in the clauses of its Articles 16 and 17, we revisited and analyzed the outputs of the SPI since 2014 (see Table S8 for details). The graphical depiction of these details in Figure 6 indicates that in the few years since its establishment, the SPI has substantially addressed various clauses under Articles 16 and 17. Gaps are nevertheless evident, while various procedural and structural limitations in the functions of the SPI hinder or prevent its full effectiveness in implementing the Articles and thus the scientific fundament of the Convention. We provide a rationale for certain limitations of the SPI's impact and how they might be effectively addressed (see Table S8).

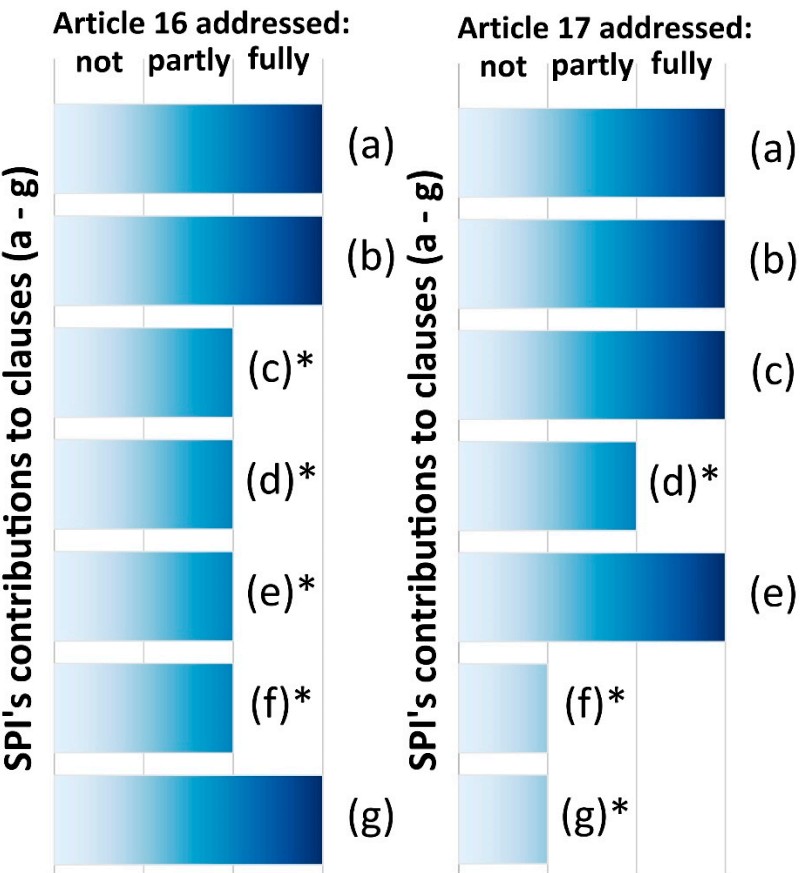

**Figure 6.** The scientific and technical impact of SPI on the various clauses (**a**–**g**) under Article 16 "Scientific and Technical Cooperation" and Article 17 "Research and Development" has been categorized to show a trend from not addressed at all, to partially addressed, to fully addressed. The smooth color progression to darker shades highlights the trend in the impact of SPI's work for each clause. The asterisk after some clauses indicates that in Table S8 we provide a rationale for certain limitations of SPI's impact and how they might be effectively addressed.

## 4. Discussion

Addressing our first objective to assess the SPI's impact on the UNCCD process, the findings presented in Section 3 show that the SPI performed very strongly in specific areas. Just over a year after its inception, the SPI's scientific work had a direct effect on a quarter of the decisions at UNCCD COP.12 in 2015, with an upward trend leading to an impact of the SPI's work on approximately a third of the decisions at UNCCD COP.14 in 2019 (Figure 2 and Table S3a–c Supplementary Materials). Analysis of all decisions since COP.13 suggests that SPI's scientific contributions have been implemented as a matter of course in various other areas of UNCCD work over the years, without SPI being explicitly cited as a scientific source (e.g., to sustainable land management to the LDN response hierarchy). This, along with the uptake of SPI work in other fora (e.g., by the G20 Environment Ministerial meeting in 2020), should be analyzed in more detail to further capture impact, potentially using automated text mining methods [42].

Regarding direct policy guidance, evidence points at SPI's scientific contributions having a significant political and strategic impact on the UNCCD process, and its international authority and visibility in addressing SDG 15. The SPI's work is particularly evident in efforts addressing Target 15.3 on LDN, the operationalization of which is inextricably linked to efforts to achieve other SDG targets [43].

The assessment of the 98 national target setting reports shows that countries are directly or indirectly using elements of the Conceptual Framework for LDN, indicating that national implementation is based on scientific evidence provided by the SPI. However,

gaps in national reports indicate that science-based capacity-building measures for operationalizing LDN need to be systematically developed by the SPI in collaboration with the GM, particularly concerning an enabling environment for achieving and sustaining LDN, elements of which have been identified by the SPI [39].

Evidence of the SPI's engagement with other organizations and the use of its products by professionals beyond those involved in the SPI's coordination activities has significantly advanced the UNCCD's authority on SDG Target 15.3 among its strategic partners and beyond, and contributed to the forging of new partnerships. SPI's contributions have raised the scientific profile of the UNCCD with other land-related UN agencies and international scientific bodies. This has occurred, for instance, through work that identified and harmonized indicators to monitor the status of land (e.g., measuring progress towards the goals and expected impacts of the UNCCD Strategic Framework 2018–2030) [21]. This work enhances the potential for synergies in addressing desertification, land degradation, drought, and land-related SDGs. Furthermore, Decision 7 of the 13th COP directly links the Conceptual Framework for LDN to the UNCCD 2018–2030 Strategic Framework. Parties to the UNCCD, therefore, plainly foresee a role for the SPI in supporting the implementation of the UNCCD's 2018–2030 Strategic Framework.

In the context of impact on UNCCD COP decisions and strategic partnership-building, evidence gathered points to SPI scientific products having an impact in the implementation of the UNCCD. However, assessing this impact on UNCCD-relevant processes and target groups beyond the direct scope of the Convention proved difficult in the absence of systematic monitoring. We, therefore, anticipate an under-estimation of the SPI's impact. Better monitoring is needed to assess the utilization of SPI products, which could include recording download frequencies of SPI technical reports and policy briefs, or systematic recording of citations of SPI products in journals, guidelines, books, or other peer-reviewed publications.

Similarly, we found that there is no central systematic documentation of peer-reviewed papers developed by SPI members during their term, even when members have later published journal articles linked to the SPI's technical reports. More recently, however, there has been some effort in gathering this information [44]. This misses additional estimation of impact and potentially significant uptake in the longer term. For example, the peer-reviewed paper discussing the SPI's Conceptual Framework for LDN [19] has been cited 290 times as of 28 February 2022, since its publication in 2018, with 35 policy citations, and 17 tweets according to Google Scholar and the publishing journal's citation metrics. However, citations of other technical reports (not published in peer-reviewed journals) may be limited due to lower visibility (e.g., search engines for academic and scientific research such as Web of Knowledge may not capture gray literature such as technical reports of UN Agencies and Conventions). Additional human and financial resources for outreach (e.g., via social media) and systematic monitoring and evaluation of the SPI's contributions to other UNCCD-relevant organizations and regional science-based networks are needed to better track impact.

Our findings show that the SPI has developed lasting communication pathways with other science–policy interfaces, such as the IPCC, IPBES and ITPS, leading to higher visibility of the UNCCD's custodianship of SDG target 15.3, which is based on science on desertification, land degradation and drought. Reciprocal consideration and citation of the reports emerging from the different science–policy interfaces could be supported by further formalizing the partnerships between these bodies (such as through a memorandum of cooperation) and would potentially enhance complementary work that could enable synergies in domestic actions on land degradation, desertification, biodiversity loss and climate change.

Analyzing experiences from IPCC and IPBES communication and engagement strategies [45,46] could provide useful insights for monitoring the "societal relevance" of SPI outputs in the UNCCD-context (see Table S2), particularly in advance of, during, and in the aftermath of SPI-relevant UNCCD meetings (World Day to Combat Desertification

and Drought, UNCCD Science Days, COPs, CRICs, etc.), informing similar social media outreach for SPI products. IPCC and the IPBES communication strategies show that a communication plan for each report is prepared well in advance of its publication. Such efforts would help to improve visibility of SPI products, promoting scientific communication between the SPI, UNCCD Parties, and strategic partners. This would help to ensure that the SPI's work is not viewed in isolation, and that while it ensures continued progress on key UNCCD scientific requirements, it also supports the long-term development of strategic scientific partnerships, fosters synergies in actions on LDN across levels, and enables resource efficiency and effectiveness, particularly for actions targeting land-related SDGs.

Downloads of SPI technical reports and policy briefs, curtain-raisers, and associated peer-reviewed publications emerging from SPI work may also be indicative not just of engagement of other scientists with the SPI's products but also that of society more broadly. It should be assumed that the SPI will produce more products over the years, resulting in peer-reviewed publications that will be used by other external professionals, and which should increase the visibility of UNCCD scientific expertise in the fields of desertification, land degradation, drought and sustainable land management. The systematic tracking of SPI citations, references in public media, including TV/radio/blog/tweets, and podcasts related to science nevertheless requires human and financial resources. Both IPCC and IPBES use the tool Meltwater [47] for monitoring 'traditional media', using extensive keywords and Boolean search operators. IPBES searches daily in more than 100 languages and reports to the Executive Secretary and the Bureau of IPBES roughly every six months, engaging a consultant to support this work during larger events and launches (Personal Communication, Robert Spaull, Head of Communications of the IPBES, April 2021).

Our second objective was to evaluate whether the SPI contributes to the achievement of the objectives of the Convention outlined in Articles 16 and 17. The findings highlight that the SPI has made numerous contributions but faces procedural and structural limitations.

A lack of focus on drought is a major shortcoming in the UNCCD implementation process since its inception in 1994. The SPI third biennial work program 2018–2019 addressed this gap, leading to the development of a new Drought-Sensitive Land Management approach. However, it is noteworthy that Article 17(g) on enhancing the availability of water resources in affected areas has not yet been addressed in the SPI's biennial work programs. Addressing this would conclude an important issue that is central to the Convention and could usefully draw upon collaborations with existing strategic partners.

Regarding Article 16(d) (on making full use of the expertise of competent intergovernmental and non-governmental organizations), systematic monitoring and recording of the SPI's contributions beyond reporting to the UNCCD on the objectives and coordination activities under its work programs is insufficiently documented to enable a full assessment of the SPI's impact on other organizations. Nevertheless, the demonstrable impact of the SPI on fostering the development of UNCCD strategic partnerships (Tables S4a–c, S5 and S7a–c) already raised the profile of the UNCCD as promoting scientific collaboration and the recognition of the growing scientific authority of its bodies in addressing desertification, land degradation and drought and sustainable land management. Resources need to be allocated to monitor and document the use of SPI products by UNCCD strategic partners. This is particularly important for Article 16(e), which calls for full weight to be given to the collection, analysis and exchange of socio-economic data, and their integration with physical and biological data. A further challenge for Article 16(e) is that the SPI has not had activities in its biennial work program that clearly call for consideration of the business sector: the aspect on how production systems and consumption clearly relate to desertification, land degradation and drought may need specific attention towards achieving LDN.

Article 16(f), which calls for exchange and to make information fully, openly, and promptly available from all public and available resources relevant to combating desertification and mitigation the effects of drought, highlights further structural constraints. The SPI's outreach capacity needs to be enhanced and institutionalized within the UNCCD Secretariat to further raise the global profile and importance of the UNCCD, including

the encouragement of relevant science to close knowledge gaps towards LDN. This will require additional human and financial resources. Based on analysis of the IPCC and IPBES systematic approaches, a UNCCD SPI dissemination and communication strategy should be developed well in advance of the publication of an SPI report and be an element of the approval phase of an SPI work program. Analysis of media processes developed for IPBES or IPCC reports, including pre-approval briefings (e.g., the preview introducing IPBES' 2019 Global Assessment Report on Biodiversity and Ecosystem Services) [48], and media training, offers valuable insight into proven measures used with other science–policy interfaces. These should be used to support SPI members, and media and outreach activities, following the release of an SPI report or product.

Clause 17(d) on developing and strengthening national, subregional, and regional research capacities, including local capacity development, and the strengthening of appropriate capacities, especially in countries with a weak research base, is not yet optimally covered by the SPI. Advancing UNCCD implementation to address this requires the interplay of all three units of the modular mechanism initially proposed in 2013 to provide scientific advice to UNCCD COPs [12,13]. While the COP resolved to establish the SPI, the recommendation for establishing Regional Science and Technology Hubs in each UNCCD region (bringing together existing regional scientific networks to collect and synthesize regional knowledge on desertification, land degradation and drought), to interact with the SPI is yet to be enacted. A strategy and actions towards the implementation of these regional Hubs could thus accelerate progress towards Clause 17(d) and could be effectively operationalized through UNCCD bodies and its Regional Annexes, enabling their formation according to specific regional needs and circumstances (see also [14]).

## 5. Conclusions

Decisions of the UNCCD COPs since 2015 have progressively integrated SPI's scientific contributions into a broad range of UNCCD deliberations and concrete actions at local, national, and international levels. Within just a few years of its existence, the SPI has emerged as a key service provider, delivering specifically requested science to the UNCCD. The SPI work programs show a progressive trend towards ensuring that the work under each work program is linked, which has strengthened the SPI's impact in terms of its contributions to COP decisions. However, comparison of the SPI's products with Articles 16 and 17 of the Convention also reveals shortcomings. Key issues are attributed to limited or insecure resourcing of the SPI's work and its non-formal status within the UNCCD. These challenges significantly compromise the effectiveness of the SPI and implementation of the UNCCD because they limit scientific collaboration with other science-based organizations, while the not-yet-formalized status within the UNCCD weakens the position of the Convention in its work to exert its authority on desertification, land degradation and drought.

Progress gaps in advancing clauses under Articles 16 and 17 should be addressed by revisiting the modular mechanism, proposed in 2013, to provide scientific advice to the UNCCD. At that time, the establishment of Regional Science and Technology Hubs was proposed to bring together existing scientific networks in each UNCCD region and to interact with the SPI. These Hubs were perceived as catalyzers of research addressing regional and sub-regional needs. Overall, it would fundamentally improve the grounding of UNCCD-relevant science and research evidence emerging from the SPI, boosting the UNCCD's regional efforts to attain and maintain LDN.

Further, the SPI needs to be legally enshrined as a permanent UNCCD science–policy body. This would facilitate its resourcing, ensuring its continued eligibility for funding through the Trust Fund. Such changes will be critical to ensure better alignment between the availability of resources, the functions and work of the SPI, and the exploration of new opportunities for collaboration. This is especially necessary and increasingly urgent in the run-up to the deadline for implementation of the 2030 Agenda and other land-related actions in the international decade on ecosystem restoration, particularly if the UNCCD is to retain its role as the global authority on land degradation, desertification, and drought.

**Supplementary Materials:** The following supporting information can be downloaded at: https://www.mdpi.com/article/10.3390/land11040568/s1, List S1: List of acronyms used in the paper; Figure S1: Flow chart showing how science-based evidence provided by SPI informs the UNCCD through its structure; Table S1: Criteria to identify existing relevant frameworks, linked to pertinent articles of the UNCCD; Table S2: Framework applied to assess SPI processes and products in the UNCCD context; Table S3a: Detailed description of SPI-informed decisions of the UNCCD conference of the Parties at its 12th session (COP 12)—based on the implementation of the SPI work program 2014–2015; Table S3b: Detailed description of SPI-informed CST recommendations which contributed to UNCCD decisions at UNCCD COP 13—based on the implementation of the SPI work program 2016–2017; Table S3c: Detailed description of SPI-informed CST Recommendations which contributed to UNCCD Decisions at UNCCD COP 14—based on the implementation of the SPI work program 2018–2019; Table S4a: Demonstrable impacts of the SPI in furthering UNCCD Strategic Partnership-Building with international organizations, including science–policy bodies under its Coordination Activities 2014–2015; Table S4b: Major aspects on demonstrable impacts of the SPI in furthering UNCCD Strategic Partnership-Building with international organizations, including science–policy bodies under its Coordination Activities 2016–2017; Table S4c: Major aspects on demonstrable impacts of the SPI in furthering UNCCD Strategic Partnership-Building with international organizations, including science–policy bodies under its Coordination Activities 2018–2019; Table S5: Demonstrable utilization of SPI products by UNCCD Parties and by professionals in the UN policy arena; Table S6a: Review of the UNCCD GM national target setting reports against direct and indirect criteria derived from the Conceptual Framework for LDN to analyze whether the scientific evidence provided by the SPI supports the national targets (the asterisk indicates that additional information is included in Table S6b on particular coding used in the indirect criteria section); Table S7a: SPI's demonstrable engagement with other organizations and the utilization of its products by professionals beyond its coordination activities that supported the visibility of UNCCD as an authority on science on DLDD in the biennium 2014–2015; Table S7b: SPI's demonstrable engagement with other organizations and the utilization of its products by professionals beyond its coordination activities that supported the visibility of UNCCD as an authority on science on DLDD in the biennium 2016–2017; Table S7c: SPI's demonstrable engagement with other organizations and the utilization of its products by professionals beyond its coordination activities that supported the visibility of UNCCD as an authority on science on DLDD in the biennium 2018–2019; Table S8: Evaluating SPI's contribution to the achievement of the objectives of the Convention outlined in Articles 16 and 17—including our analyses of limitations in current SPI contributions, processes or structures.

**Author Contributions:** Conceptualization, M.A.-S., G.M., L.C.S. and N.N.B.; methodology, M.A.-S., G.M., L.C.S. and N.N.B.; validation, M.A.-S., G.M., L.C.S., N.N.B., G.K. and J.-L.C.; formal analysis, M.A.-S., G.M., L.C.S., N.N.B., J.-L.C. and G.K.; investigation, M.A.-S., G.M., J.-L.C., N.N.B. and G.K.; original draft preparation, writing, M.A.-S., L.C.S. and G.M.; review and editing, L.C.S., G.M. and N.N.B.; visualization, N.N.B., G.M. and M.A.-S.; supervision, M.A.-S. All authors have read and agreed to the published version of the manuscript.

**Funding:** This research received no external funding.

**Institutional Review Board Statement:** Not applicable.

**Informed Consent Statement:** Not applicable.

**Data Availability Statement:** All data used for analysis are publicly available and data sources are indicated. Further information is included in the Supplementary Materials.

**Acknowledgments:** We wish to extend our special thanks to the UNCCD secretariat, specifically Barron Joseph Orr, Sara Minelli, Satu Ravola, Yukie Hori, Wagaki Wischnewski and Katya Arapnakova, for their valuable technical assistance in tracking SPI relevant actions, and for their personal communications that clarified our queries in relation to selected SPI-related processes. We would also like to thank the Global Mechanism of the UNCCD, specifically Cathrine Mutambirwa, Pedro Lara Almuedo and Andjela Vragovic, who provided essential support that made the analysis of the LDN target setting reports possible. We express our appreciation to the DLR Projektträger, Berlin, Germany for covering the publication costs.

**Conflicts of Interest:** The authors declare no conflict of interest. The funders had no role in the design of the study; in the collection, analyses, or interpretation of data; in the writing of the manuscript, or in the decision to publish the results.

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
