# Peer review of "Assessing the Impact of Science in the Implementation of the United Nations Convention to Combat Desertification"

_land, doi:10.3390/land11040568_

Round 1
Reviewer 1 Report
An excellent paper making a strong case for the stronger formalisation of science in this vital UN Convention. Well written and argued. No changes required in my opinion. Publish as-is.
Author Response
We are delighted that you enjoyed our paper and have recommended its publication with no changes.
Reviewer 2 Report
OUTLINE AND GENERAL APPRECIATION
This paper assesses the impact of the outputs generated by the science-policy interface (SPI) on the decisions of the UNCCD. It shows, in the light of various documents, the growing importance of this body. The work is very well referenced and the supplementary material helps.
GRAMMAR AND LANGUAGE
The document is very well written. The English is perfect, so there is nothing to complain about in this area.
SOME SUGGESTIONS
Despite good writing and structuring, the manuscript is a bit cumbersome due to the number of acronyms used. Section 3.3 is particularly dense and could perhaps be summarised.
In my opinion there are a couple of things that could help the reader. One is a flow chart to help understand how the UNCDD is structured and how information goes from one body to another (the so-called UNCCD process). The second idea is to present a table with all the acronyms used, so that it can be consulted. Sometimes I have had to go back to look up what certain acronyms stand for. Since the text often strings together several acronyms, it is easy to lose the thread in one of the sections.
One of the few questions I have when I read it is exactly which SPI products it refers to. I understand that they are reports or guidelines (e.g. Good Practice Guidance for SDG Indicator 15.3.1). In table S5 these products are presented under a numbering that I can't see what it means: 1.1, 1.2, etc.
Author Response
Thank you for your positive feedback on our paper and your appreciation of the work that went into both the provision of evidence in the main body of the paper and the supplementary materials.
We agree with your comment about the number of acronyms so have simplified section 3.3, reducing the acronyms to only the well known ones and writing others like CF LDN (conceptual framework for LDN) out in full where possible (not in the Figures though, to avoid them becoming too cluttered). The language has been made more concise where possible too.
A flow chart has been created and placed in the supplementary information to show how information goes from one component to another. We also specifically show where the SPI plays a key role. This is labelled Figure S1.
We have also aided the reader by including a table of acronyms and abbreviations at the very start of the supplementary information (called “List of acronyms and abbreviations). Reference to this and the flow chart has been made in the paper in lines 86-69.
We have clarified Table S5 with a note after the table title: Products are numbered under each biennium. Numbers do not relate to specific types of product. We have also labelled the column Product number now. This was done to help improve the visibility of the quantity of SPI products as they tended to become less visible within the UNCCD process.
The longer explanation for our approach (which we haven’t put in the paper but is possibly useful to outline in response to your comment) is that SPI products are essentially all outputs resulting from SPI’s biennial work programmes. During the first work programme (2014-2015), policy-oriented proposals were developed that fed directly into CST documents that directly influenced COP decisions. As a result, they became less visible as stand-alone SPI products. During the successive 2016-2017 and 2018-2019 work programmes, the SPI produced technical reports (with the SPI’s own logo), conceptual frameworks, indicators, and decision trees. In response to Decision 18/COP.12, SPI began preparing policy briefs on the policy implications of recent developments in scientific research relevant to desertification, land degradation and drought, and land-based climate change adaptation and mitigation.
Reviewer 3 Report
I like this paper very much. I think it can be published with very minor revisions and adaptations. I would see a more 'regional' discussion about UNCCD annexes and their implications for desertification fighting in different parts of the world. For the rest, the article is very smart and well presented/illustrated. Just think to the fact that it is possible to classify the paper as 'commentary' or 'case report' because the actual classification of 'article' ( a typical scientific paper with models, results, empirical analysis of data) is likely a bit different from the content of the paper.
Author Response
We thank you for your review comments and we are very pleased that you enjoyed our paper. We are grateful that you have pointed to the crucial aspect that the relevance of the UNCCD is intrinsically linked to its implementation at the regional level. This is clearly reflected in our discussion that additional resources are required for the systematic monitoring and evaluation of the SPI’s contributions also to regional science-based networks. Based on our analyses of clause 17(d) we also argue that regional science and technology hubs in each UNCCD region are required.
To specifically address your comment, we have made reference to the UNCCD annexes. The sentence now reads (p. 15, L662-663: (new inserts indicated in bold): “A strategy and actions towards the implementation of these regional Hubs could thus accelerate progress towards Clause 17(d), and could be effectively be operationalized through UNCCD bodies which are included in its regional Annexes, to enable their formation according to specific regional circumstances and needs (see also [14]).”
We also now refer to regional annexes in lines 44-45 in the introduction where we note that: “While the UNCCD specifically addresses dryland areas, it fosters sustainable land management (SLM) across all climatic zones, while regional annexes allow these challenges to be addressed according to the specific regional contexts in which they occur“.
Regarding your feedback about the type of article, we have raised this with the editor too, as we think that while we don't take an experimental approach, what we have produced is still original empirical research and that it should be recognised as such.
The Land guidance for authors on the website indicates the following: “Articles: Original research manuscripts, which should comprise at least 15 pages. The journal considers all original research manuscripts provided that the work reports scientifically sound experiments and provides a substantial amount of new information.“
The work we have presented in this paper constitutes original research and provides a substantial amount of new information. The paper is at least 15 pages long as well, so we consider it is entirely appropriate for the paper to be recognised as a research article. Our approach follows established social science methodology and uses document analysis methods (with appropriate referencing to the social science methods literature) to justify and support our research.
We would therefore appreciate it if the journal would recognise the new empirical work within the paper as an article, as would be the case in all social science journals. We think this is important, particularly given the journal considers itself to be interdisciplinary. We realise that different approaches dominate in different disciplines, but this needs to be recognised in an interdisciplinary journal.